# A cloud-assisted key agreement protocol for the E-healthcare system

Ismail Keshta * 

Department of Computer Science and Information Systems, College of Applied Sciences, AlMaarefa University, Riyadh, Saudi Arabia

* imohamed@um.edu.sa

## Abstract

Secure data transmission is critical to cloud-based electronic healthcare (e-healthcare) systems. Upon delving into the literature, it becomes clear that several security mechanisms have been developed to ensure the security of exchanged information across patients, physicians, and the cloud server, but they do not resist vulnerabilities such as man-in-the-middle, side-channel, and session key hijacking attacks. These vulnerabilities can seriously compromise the integrity of exchanged medical records. In light of this, the article proposes a cloud-assisted key agreement protocol for the e-healthcare system to enable secure authentication for patient monitoring, enhancing mutual authentication of the participating entities and creating protected session keys for secure open-channel communication. The proposed methodology employs robust and lightweight procedures, including SHA-256 and Elliptic Curve Cryptography (ECC), while considering the challenges of implementing strong security protocols in healthcare systems, such as the need for high performance and low energy consumption. The proof of correctness and robustness of the proposed protocol has been analyzed through the Real-Or-Random (RoR) model, ProVerif verification toolkit, and pragmatic illustration, while the efficiency and efficacy were checked by measuring computation, communication, storage costs, and energy consumption. The result obtained from the security analysis demonstrated that the proposed protocol resisting man-in-the-middle, replay, DoS, traceability/tracking, desynchronization, impersonation, and side channel attacks offers key secrecy, confidentiality, integrity, and authorization. In contrast, the result depicted from the performance analysis section shows that the proposed protocol is 46.99% better in communication, 96.46% in computation, and 53.69% in energy consumption, which is inaugurating its superiority over its competitors. Hence, it is recommended for practical implementation in the real-world cloud-based e-healthcare scenario.

**Data availability statement:** All the data is avialbel within the manuscript.

**Funding:** The author(s) received no specific funding for this work.

**Competing interests:** The author declared that he has no known competing interest with any entity.

## Introduction

The e-healthcare system, a complex, networked structure, demonstrates the power of technology in revolutionizing healthcare delivery. This heterogeneous distributed system, comprising many components, tools, and technologies, collaborates to enhance patient care and health outcomes. With its diverse functions, communication protocols, and data formats, it seamlessly integrates a wide array of hardware, sensors, and software platforms. For example, the system incorporates wearable sensors, mobile devices, cloud servers, and medical databases. The geographical dispersion of the system's components confirms its versatility. For instance, wearable sensors can collect patient data at home, which can then be transmitted to a local gateway, processed on a cloud server, and accessed by a remote doctor [1]. Cloud computing gives users on-demand access to resources like storage, processing power, and apps because the underlying systems are managed and maintained by cloud service providers (CSPs); this paradigm does away with local hardware or infrastructure requirements [2]. E-healthcare-driven hospitals are reassured by this arrangement, which frees them from the inconvenience of hardware monitoring and maintenance so they can concentrate on using computer resources [3]. Public, private, hybrid, and multi-cloud systems are among the many deployment choices offered by cloud computing, each with unique advantages and factors to consider. Public clouds are accessible to many customers via third-party cloud service providers [4]. Private clouds, on the other hand, can be hosted on-site by a third-party provider or customized just for one company [5]. By combining the benefits of private and public clouds, hybrid clouds allow businesses to benefit from each of their finest features [6]. Multi-cloud settings, on the other hand, use services from several cloud providers, reducing CSP lock-in and increasing efficiency, cost-effectiveness, and dependability [7]. All things considered, cloud computing offers more flexibility, scalability, and cost reductions than traditional physical infrastructure, radically changing how patient and e-healthcare service providers access and use computer power. For instance, cloud-based electronic health records (EHRs) have made patient data more accessible and secure, providing reassurance and confidence in better-informed decisions and improved patient outcomes [8].

For the e-healthcare system services, the integration of cloud computing resources like storage, processing power, and software applications is a powerful tool. It empowers healthcare professionals to improve productivity, connection, and care quality in patient diagnoses. The trust is bolstered by the use of wearable/sensors and IoT technology, which connect with the cloud server to make remote monitoring of sensitive patient signs and health data easier. The cloud-based video conferencing technology further enhances e-healthcare services, enabling physicians to confidently conduct remote consultations with patients and medical professionals. Healthcare companies can harness the power of cloud computing to analyze vast amounts of data, find patterns, and gain insights that guide clinical judgment and enhance patient care. Providers can use cloud platforms to deploy artificial intelligence and machine learning algorithms, empowering them in infection diagnosis, patient outcome prediction, and personalized treatment plans [9]. With the aforementioned importance of the

cloud-assisted e-healthcare systems, integrating different technologies utilized by patients and physicians, avoiding large expenditures in physical infrastructure, and scaling their computer resources up or down as needed, cloud computing brings significant efficiency and time-saving benefits. This not only saves money but also eases the workload for all stakeholders, improving resource management [10]. However, the security of records in all participating entities depends on the sound service providers, their strong security protocols, and compliance guidelines to ensure safety against cyber threats, illegal access, and vulnerabilities [11].

## Motivation

The existing authentication protocols presented by different researchers from time to time (discussed in detail in the related work section of this article) for cloud-assisted e-healthcare systems either suffered from privacy, anonymity, efficiency, and design issues or failed to resist impersonation, DoS, insider, desynchronization, replay, MITM, and side-channel attacks. Some of these protocols suffer from scalability issues, complexity in implementation, and limited device compatibility challenges, while others are susceptible to specific types of attacks, such as ESL, tracking/traceability attacks, or session key hijacking issues, and some of them have maximum computation due to modular exponentiation.

Furthermore, a robust authentication of all the participating entities must be followed because the cloud-based e-healthcare infrastructure designated for patient health information is extremely sensitive and needs to be protected against abuse, breaches, and unauthorized access. The data sent between sensors, wearables, and IoT devices to the cloud server is susceptible to numerous threats, modifications, and interruptions by adversaries without secure key agreement methods. The inadequacy of current security measures is a pressing issue that demands improvement and innovation. Healthcare professionals, IT security experts, and developers play a crucial role in ensuring the security of said cloud-assisted e-healthcare system, but unfortunately, internet-connected devices (sensors, wearables, or IoT devices) are resource-constrained and lack inadequate security measures to address contemporary threats, and the said service providers occasionally mismanage their security features. It is also necessary to introduce network capabilities in these network-enabled devices, which is needed for a dynamic authentication protocol (like the protocol proposed in this article), but it is too challenging and needs careful consideration and strategic planning, which is often lacking in current cloud service providers. Considering all the aforementioned issues and challenges, there is a need to address the critical security, privacy, and efficiency problems in the exchanged patient-sensitive information by designing a cloud-assisted key agreement protocol in the e-healthcare system that can guarantee to mitigate all the weaknesses discussed above.

## Contributions

The contributions of this research are summarized in bullet points, highlighting the innovative application of advanced technologies in developing this new protocol.

- To propose a key-agreement protocol for a cloud-assisted e-healthcare system operating without passwords or user biometrics that has the potential to significantly improve security by eliminating online/offline password guessing, password breaches, and compromised biometric data by utilizing a lightweight and robust cryptographic technique called elliptic curve cryptography (ECC) and a collision-free one-way hash cryptographic function called SHA256 algorithm.

- To comprehensively analyze the proof of correctness and robustness of the proposed key agreement protocol employing the well-known and widely used Real-Or-Random (ROR) model and ProVerif simulation, confirming integrity, secrecy, confidentiality, and reachability of the patient-sensitive information exchanged in a potentially hostile environment.

- To methodologically evaluate the performance metrics of the proposed key agreement scheme using a testbed research method, confirming its effectiveness, strength, and efficacy while considering computation, communication, storage costs, and energy consumption.

 

- To comparatively analyze the proposed protocol for performance metrics in checking how well it balances with security, the often conflicting aspects of a protocol are frequently ignored in previous studies.

The remainder of the article is organized as follows: in the related works section, the review of existing schemes was discussed, and their pros and cons are presented in the form of a table; in the system architecture section, the role of each participating entity, how it will work and associated aspects were given in textual as well as diagrammatically; in the proposed key agreement scheme section, the solution is presented in three phases, including setup, registration, and authentication. The algorithmic representation has been given for the proposed scenario, emphasizing its practical implications and making the relevance and applicability of the research clear. In the security analysis section, a demonstration regarding the security of the proposed key agreement scheme has been presented both formally using the RoR model and informally through pragmatic discussion. In contrast, in the performance and comparative analysis section, an illustration of the performance metrics in terms of communication, computation, storage, and comparative analysis has been presented, and in the conclusion section, what has been done in this article for easy understandability of the readers.

## Related works

Cloud computing can offer services to many fields ranging from logistics to e-healthcare systems; however, security is still a major concern that is repeatedly noted in the cloud computing paradigm because numerous vulnerabilities are identified from time to time. In this connection, researchers from different times have proposed numerous security schemes to make the system secure and attractive for all areas, especially health care, where patient-sensitive records are publically exchanged. These proposed security schemes are solutions and the future of e-healthcare security. Lopes et al. [12] proposed a protocol to provide safe and reciprocal device authentication within the system. However, this protocol is vulnerable to a traceability attack because the identity is transmitted openly over a public network channel, through which an attacker can easily trace a legitimate user. Ayub et al. [13] introduced a lightweight authentication protocol for e-health clouds using a three-factor authentication mechanism to guarantee safe access in an IoT-based system, but it has design issues because when a legitimate user desires to update their credentials, it never changes in the succeeding entities. Khan et al. [14] securely authenticated the e-health sector utilizing blockchain technology with the addition of conventional cryptographic techniques, but efficiency is still an issue there, as the computation cost is too high and maximum bandwidth is utilized. Shariq et al. [15] have significantly contributed to cloud computing by devising radio frequency identification (RFID)-based authentication methods, but traceability has not been appropriately tackled, as the patient is traceable through the RFID serial number. Ansari et al. [16] have proposed a privacy-enabled architecture for cloud-based e-healthcare systems, further advancing the field but prone to impersonation and DoS attacks because the secret key and random numbers are too short that adversaries can easily break and reach the sensitive credentials.

Masud et al. [17] made a compelling case for the urgent shift from conventional to cloud-based healthcare. They then proposed a scheme to ensure secure access to electronic healthcare records by employing a key generation function for end-to-end encryption and efficient access to the cloud server, but they failed to examine/scrutinize the proof of correctness and robustness of their scheme. Padmaja et al. [18] addressed the security issues in the cloud computing paradigm by proposing an authentication scheme for patients to avail of medical services in an effective manner. However, due to using the MD5 encryption technique in the device authentication phase, it is inadequate, falling short of providing the robust security necessary for the e-healthcare system, as MD5 is weak against a hash collision attack. Chandrakar et al. [19] concentrated on the design of a cloud-based protocol through a hybrid cryptosystem method to monitor the e-healthcare record and protect patient privacy via mobile phones without attending the hospital. However, their protocol's performance is excessively high due to the use of bilinear mapping and XOR, making it easy to trace the patient. Deebak et al. [20] presented an intelligent service authentication architecture that uses symmetric cryptography, XOR operations, bilinear pairing, and bio-hashing. Their [20] technique used biometric data for hashing to improve mutual authentication

in cloud-based medical systems, but the communication cost is high because of the expansive execution time complexity due to modular exponentiation in paring cryptography.

Chiou et al. [21] presented a cloud-enabled e-healthcare system that offered services to patients in an efficient manner via a telemedicine system, ensuring patient privacy and unlinkability, message authentication, and lower computational costs due to modular exponentiation. However, they did not address the problems of key hijacking, server spoofing, and attacks, and doesn't preserve patient anonymity, as an adversary on their scheme easily launches an identity-guessing attack. Qadir et al. [22] employed a modular approach in which a registered patient was granted access to medical documents many hospitals do not offer. However, privacy isn't considered a serious concern, and hence patient privacy is a big issue in the model, underscoring the ethical implications of the their systems. Okikiola et al. [23] focused on identifying an insider threat in the cloud-based system and proposed a methodology that uses watermarking extraction and logging methods employing symmetric encryption/decryption and watermark extraction techniques to identify fraudulent activity in the telemedical information server. However, their model did not take into account record alteration. Benil et al. [24] suggested an elliptic curve aggregate certificateless signing technique for data integrity and secrecy using blockchain technology, but the performance of their hybrid system is too high for practical applications because of the elliptic curve discrete algorithmic problem (ECDLP) and exchange of public key each time require high bandwidth.

Alqarni et al. [25] presented a lightweight authentication system for the resource-constrained healthcare setting and showed that their plan would effectively provide services to deployed devices; however, the second message, which is sent from the gateway node to the sensor node, has an identity in plain text format, which an attacker may discover and use to initiate an insider, tracking threats and create privacy issues to the system. Abbasi et al. [26] offered security architecture for the healthcare system that would allow all involved entities to access associated medical facilities efficiently. Notably, despite its security, this method ensures the reliability of two-party authentication, as the cloud-assisted e-health system needs to be effective and privately deal with every patient and guarantee the confidentiality and security of electronic medical records, which is unfortunately missed in [26]. Resolving security and privacy concerns, [27] said that the cloud-assisted electronic healthcare system should be designed to retrieve electronic medical records as quickly as possible to attract more people towards cloud-equipped healthcare systems. However, their strategy lacks a privacy security feature because the patient preferences, location, and coordinates are easily traceable due to the easy launch of an identity spoofing attack on their scheme. The authors of [28] argued that a cloud-based system is an attractive solution for efficiently exchanging electronic medical records and protecting everyone's privacy. However, privacy is still an issue in the system due to the inability to adequately protect user credentials while exchanging them over an open channel. The authors in [29] proposed a fog-assisted health data sharing technique, a method that utilizes edge computing to process data closer to the source, which is a safe and efficient method and increases the effectiveness and privacy of patient-sensitive information; however, they failed to verify the dynamic nature of their scheme, because all the credentials are once generated and utilized throughout the process. The author [30] proposed a symmetric-based encryption method for cloud-assisted smart healthcare systems for straightforward, secure data sharing but found it vulnerable to insider, masquerade, and man-in-the-middle attacks because an adversary can easily pick the transmission of the parameter over a public network channel and later act as a malicious user and launches the said attacks. The summary of the critical literature review is demonstrated in Table 1.

In conclusion, various protocols and strategies have been developed for cloud-based e-healthcare to ensure patient-sensitive records' confidentiality, security, and efficiency, maintain privacy, and securely exchange data amongst all participating entities. These schemes either suffer from impersonation, masquerade, side-channel, traceability attacks having design flaws, or heavyweight due to weather using heavyweight cryptographic algorithms or completed in three to four round trips. Also, these schemes lack privacy, anonymity, or key secrecy features or have outdated data transmission flaws. To cope with these issues and challenges, this research proposes a cloud-assisted key agreement technique for a healthcare system to securely monitor a patient remotely. This study is grounded in the premise that our proposed

**Table 1. Summary of the critical literature review.**

| Ref | Technique | Pros | Cons |
|-----|-----------|------|------|
| [31] | Asymmetric cryptography | • Lightweight scheme<br>• Mitigated patient privacy | • Outdated data transmission flaw<br>• Side channel attack<br>• Replay attack |
| [32] | Hybrid cryptosystem and Massive Machine Type Communication (mMTC) | • 5G-enabled mMTC<br>• Mutual authentication<br>• Session key agreement<br>• identity anonymity | • No feasible for resource constrained IoT<br>• Privacy issue<br>• ELS attack |
| [33] | ASCON Algorithm, AES, and XOR Biometric Fuzzy Extractor | • Robust Scheme<br>• Scyther simulation<br>• Safe side channel attack | • Not feasible for resource constrained IoT applications<br>• Key secrecy issue |
| [34] | SHA, XOR and Symmetric Encryption/Decryption | • Correctness confirmed<br>• AVISPA simulation | • Scalability is a key factor in cloud platform, which lack of performance metric analysis<br>• Tracking Attack<br>• Collision Attack |
| [35] | HL7.FHIR and DICOM | • Easy accessing<br>• Effectively managed<br>• Cloud platform is robustly accessed through their model | • Correctness conformation has not been made<br>• Side channel attack<br>• Desynchronization attack |
| [36] | PUF and RFID along with SHA and XOR operations | • Fingerprint act is a key<br>• Safe against physical attack<br>• RFID is easily manageable | • Privacy issue<br>• Record sharing with legitimate person is not possible/difficult<br>• Insider threat |
| [37] | ECC and SHA512 | • Telemedicine provide services to physician<br>• Cloud server provide services to patient<br>• Excellent approach | • Key escrow issue<br>• Forgery attack<br>• Privileged insider attack |
| [38] | ECC and SHA256 | • Robust mechanism<br>• Integrating many technology for sensitive task<br>• Foolproof analysis | • Running time key establishment is not feasible for resource constrained tiny sensors<br>• Session key hijacking issue |

method, with its focus on maintaining patient privacy, has the potential to significantly impact the e-healthcare industry, providing essential services to patients while supporting healthcare providers and medical professionals.

## System architecture

To utilize the e-healthcare system, paramedical staff (including nurses, doctors, physicians, and patients) must first register with the cloud-based platform, in which the registration-related credentials of patient sensors/wearables or IoT devices and paramedical staff mobile devices are stored on a cloud server. This cloud server plays a pivotal role in the system's architecture, providing a secure environment for all data, including patient and paramedical staff information. Subsequently, patients can contact physicians through the cloud server to receive diagnoses and treatment plans, which minimizes the amount of information transmitted. This enables physicians to swiftly evaluate patients' physical conditions and provide more effective treatment, ultimately enhancing productivity. The system architecture for the proposed cloud-assisted key agreement scheme, detailed in this section, consists of three entities: the patient, equipped with either internal sensors or wearables for real-time psychological data collection; the cloud server, a central entity responsible for networking and setup, providing a secure environment for both patient and paramedical staff; and the paramedical staff, which can be a physician for real-time patient examination and treatment or a nurse for care. The real-time physiological data is transmitted to the cloud-assisted e-healthcare system, allowing physicians to efficiently access it on their own devices through a cloud data center, as shown in Fig 1.

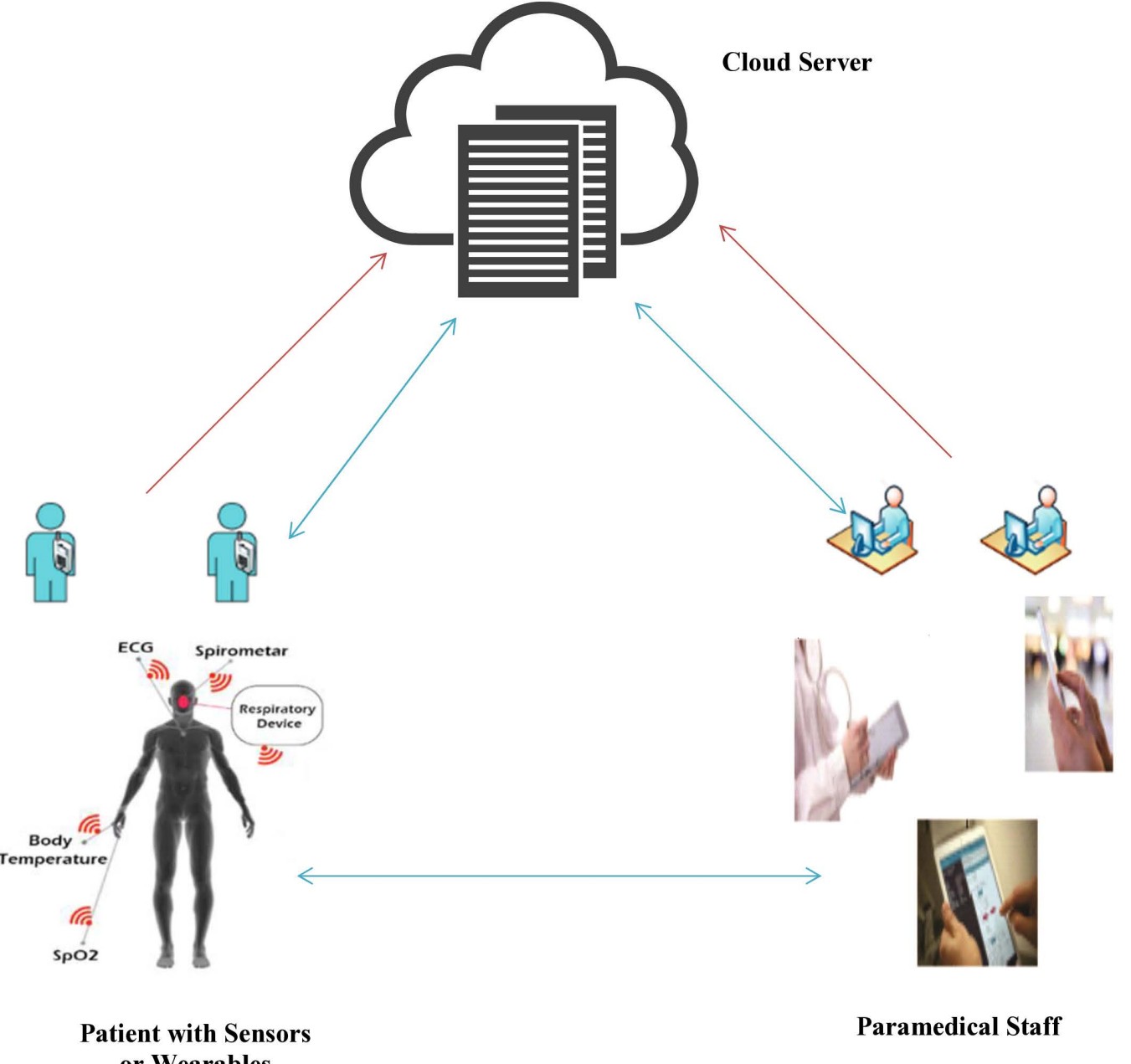

**Fig 1. System Architecture.**

## Patient sensors/wearables (P)

Biosensors, robust tools that merge biology and technology, enable real-time, precise, and portable detection of various analysts. They have revolutionized illness management, patient monitoring, and diagnostics in the medical field. With continuous advancements in data analytics, nanotechnology, and materials science, biosensors are set to become even more critical in the medical field and beyond. By combining a biological component with a physicochemical detector, these sensors assess the presence or quantity of certain compounds, often in real time. They are widely used in diverse fields,

including biotechnology, food safety, environmental monitoring, and healthcare. Biosensors play a crucial role in tracking physiological indicators, identifying illnesses, and managing long-term disorders in healthcare [39].

The embedded sensors in the human body are capable of continuous monitoring, collecting and transmitting vital health-related data (physiological vitals) to a cloud data center (server), serving as a key component in healthcare technology. For instance, temperature sensors monitor the body's standard temperature, while oxygen saturation sensors track blood oxygen levels. Visual sensors evaluate eyesight, and pressure sensors measure the duration of a patient's breathing or the stress exerted on the central nervous system (CNS) and lower jaw. Additionally, EEG, ECG, and MRI sensors assess the heart and other bodily functions, while ventilator sensors ensure a continuous oxygen supply to patients. These sensors and wearables provide a constant stream of data, ensuring that patient care is continuous and comprehensive [40].

### Mobile device (PM)

In the realm of e-healthcare systems, integrating mobile devices with physicians is a transformative strategy that empowers healthcare providers. It enhances healthcare delivery through improved data access, real-time communication, and remote monitoring. This integration allows healthcare providers to improve patient results by leveraging the widespread adoption and advanced capabilities of smartphones, tablets, and other mobile devices. They facilitate physicians for secure data access, real-time monitoring, and enhanced communication, all contributing to improved patient care and operational efficiency.

### Cloud server (CS)

A cloud data center/server is an integral part of the e-healthcare system, facilitating all the associated entities (wearables, sensors, IoT, mobile devices, etc.) for storing, managing, and processing healthcare-related matters and applications. It gives patients and healthcare personnel real-time access to affordable services, enabling both groups to monitor health-related concerns instantly. The efficiency of cloud data centers in managing and processing healthcare-related matters ensures that these systems can handle a large volume of data with ease and speed. Additionally, a cloud server offers scalability, privacy, and security to both patient and paramedical staff by playing a crucial role in the e-healthcare ecosystem, delivering timely medical assistance, performing life-saving procedures, and transporting patients to medical facilities. It is trained to provide emergency medical care and support healthcare professionals effectively, provide networking facilities and run the resources efficiently and effectively.

### Proposed protocol

This article section presents the proposed cloud-assisted key agreement protocol for e-healthcare. The proposed key agreement protocol consisted of initialization, registration, and key agreement phases, which are explained below. The symbols and notations used for the design of the protocol are presented in Table 2.

**Table 2. Symbols and Their Description.**

| Notation | Meaning | Notation | Meaning |
|---|---|---|---|
| C | Cloud Server | a, b | Point over curve |
| P | Patient | s | Secret Key |
| PM | Paramedical Staff | h(.) | Hash function |
| N | Random numbers | $K_P$ | Public Key |
| $ID_P$ | Patient Identity | $ID_{PM}$ | Paramedical Staff Identity |
| \|\| | Concatenation Function | Δ | Matching Function |

## Initialization phase

In this phase of the protocol, the cloud server chooses an integer secret number $\mu \in Z_q^*$ over curve point $(a, b) \subset q$, a collision-free one-way hash function $h(.)$, private key $s$, and produces produced $\{\mu, h(.), E_q(a, b)\}$ public parameters, and $s$ as the secret key.

## Registration phase

The registration phase is accomplished in two sub-phases: patient registration (sensor/wearable registration) and physician registration (mobile device registration). The steps of computations performed while registering these entities are explained as follows:

**Patient sensor/wearable registration.** First, the patient selects a unique identity $ID_P$, and the system picks a random number $N \in Z_q^*$ and calculates $HPID_P = h(ID_P \| N)$. The patient side terminal sends $\{HID_P, ID_P\}$ to the cloud server. When the cloud server receives the $\{HPID_P, ID_P\}$ message, it also selects a random number $N \in Z_q^*$, calculates public key $K_P = N.\mu$ and $H_P = h(ID_P \| K_P)$, $X_P = (N \oplus H_P) \| s$. The system ensures the security of the communication by sending $\{s, X_P, H_P, K_P\}$ back to the patient terminal, where it is stored in its memory, as shown in Fig 2.

**Physician mobile device registration.** Secondly, the paramedical staff/doctor or nurse selects a unique identity $ID_{PM}$; the app installed in their portable device picks a random number $N \in Z_q^*$ and calculates $HPID_{PM} = h(ID_{PM} \| N)$. This secure calculation is then sent, along with the original IDPM, to the cloud server. Upon receiving the $\{HPID_{PM}, ID_{PM}\}$ message, the cloud server also selects a random number $N \in Z_q^*$, calculates public key $K_P = N.\mu$ and $H_{PM} = h(ID_P \| K_P)$, $X_{PM} = (N \oplus H_{pM}) \| s$, and sends $\{s, X_{PM}, H_{PM}, K_P\}$ back. This data is then stored in the cloud server's memory, as shown in Fig 3.

## Authentication and key agreement phase

This is the most crucial phase of the proposed protocol, in which the three participating entities agreed on a single key to alter secure communication. This phase, with its intricate and vital computation steps, is the backbone of the protocol.

**Step 01:** The patient selects identity, chooses a random number, calculates $X_{PM} = (N.\mu) \| T_1$ and sends $\{K_P, HPID_{PM}, X_{PM}, T_1\}$ towards the physician. The physician, after verifying the time $T_1 - T_c \le \Delta T$, calculates $K_P = X_{PM} \oplus h(HPID_{PM} \| K_{PM})$, $Y_P = (X_{P-M} \| (X_{PM} \oplus N) \| K_P)$, and sends $\{HPID_P, K_P, Y_P, T_2\}$ message towards the cloud server, a crucial component in this distributed protocol.

**Step 02:** The cloud server when receiving $\{HPID_P, K_P, Y_P, T_2\}$ parameters, checks $T_2 - T_c \le \Delta T$, calculates $K_P^* = X_{PM} \oplus h(HPID_{PM} \| K_{PM})$, confirm $K_P^* ?= K_P$, compute $Y_P^* = (X_{PM} \| (X_{PM} \oplus N) \| K_P)$, verify $Y_P^* ?= Y_P$ calculate the session key $SK = h(Y_P \| K_P)$, $R_P = h(SK \| X_{PM})$, and sends $\{K_P^*, Y_P^*, R_P, T_3\}$ message back to the physician/doctor/nurse.

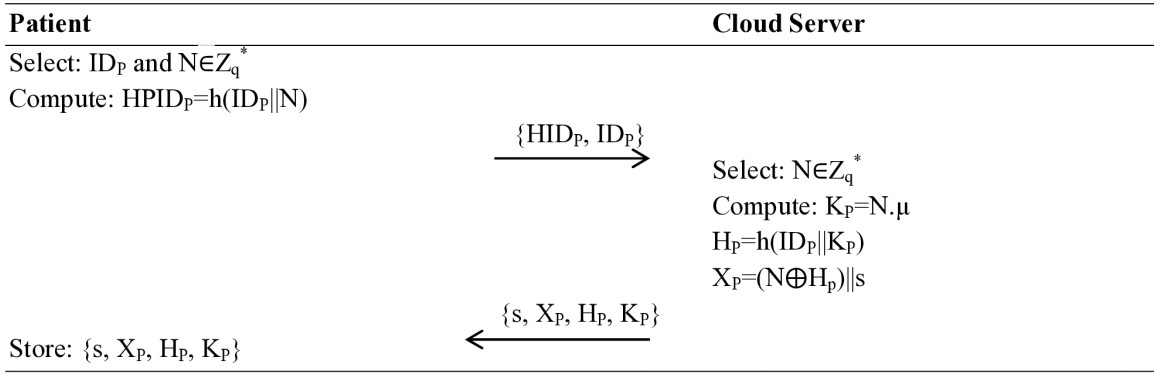

| Patient | Cloud Server |
|---|---|
| Select: $ID_P$ and $N \in Z_q^*$ | |
| Compute: $HPID_P = h(ID_P \| N)$ | |
| $\xrightarrow{\{HID_P, ID_P\}}$ | |
| | Select: $N \in Z_q^*$ |
| | Compute: $K_P = N.\mu$ |
| | $H_P = h(ID_P \| K_P)$ |
| | $X_P = (N \oplus H_P) \| s$ |
| $\xleftarrow{\{s, X_P, H_P, K_P\}}$ | |
| Store: $\{s, X_P, H_P, K_P\}$ | |

**Fig 2. Sensor/Wearable Registration.**

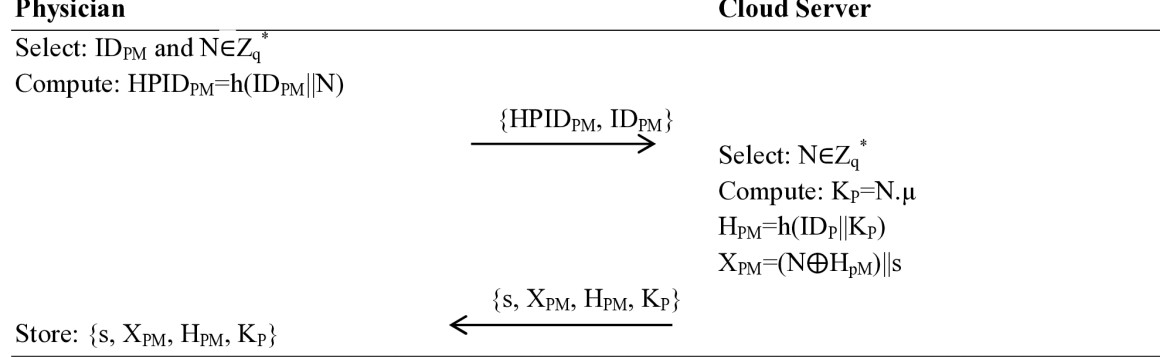

| Physician | Cloud Server |
|---|---|
| Select: $ID_{PM}$ and $N \in Z_q^*$ | |
| Compute: $HPID_{PM}=h(ID_{PM}\|N)$ | |
| | |

**Fig 3. Mobile Device Registration.**

**Step 03:** They too verify Check: $T_3-T_c \leq \Delta T$, calculates $K_P^{**}=X_{PM} \oplus h(HPID_{PM}\|X_{PM})$, verifies $K_P^{**}?=K_P^*$ compute $Y_P^{**}=(X_{PM}\|(X_{PM} \oplus N)\|K_P)$, validate $Y_P^{**}?=Y_P^*$, calculates the session key $SK=h(Y_P\|K_P)$, $R_P^*=h(SK\|X_{PM})$, confirms $R_P^*?=R_P$ and sends $\{K_P^{**}, Y_P^{**}, R_P^*, T_4\}$ message to the patient.

**Step 04:** The patient when receiving $\{K_P^{**}, Y_P^{**}, R_P^*, T_4\}$ message, validates Check: $T_4-T_c \leq \Delta T$, computes $K_P^{***}=X_{PM} \oplus h(HPID_{PM}\|X_{PM})$, confirm $K_P^{***}?=K_P^{**}$ calculates $Y_P^{***}=(X_{PM}\|(X_{PM} \oplus N)\|K_P)$, verifies $Y_P^{***}?=Y_P^{**}$ computes the session key $SK=h(Y_P\|K_P)$, $R_P^{**}=h(SK\|X_{PM})$, verifies $R_P^{**}?=R_P^*$ and keeps SK as session key as shown in Fig 4.

A thorough algorithmic description of the suggested key agreement protocol has been explicitly established, and the Python programming language has been used to implement it. A careful evaluation of the cryptographic key produced by this implementation has shown that the protocol is resilient to various security risks, such as insider threats, forgery attacks, and session key hijacking attacks. The protocol's remarkable effectiveness in addressing these vulnerabilities underscores its suitability for safe communication in the cloud computing paradigm. The session secret key generated is shown below while the algorithmic representation is shown in Fig 5.

## Security analysis

This section thoroughly analyzes the proposed key agreement protocol. We use both formal techniques, such as the Real-Or-Random model [41,42] and [43], and ProVerif [44] validation, and informal methods, including discussion and illustrations. This comprehensive analysis demonstrates the correctness of the proposed cloud-assisted protocol, which we describe in detail below.

## RoR analysis

The RoR model is applied to the suggested technique in this study. In cryptography, ROR [41]-[42] is an idealized hypothesis model used to evaluate the security of algorithms and protocols. Formal models and strict mathematical reasoning confirm the protocol's security. P, PM, and CS are the three entities that actively participate with one another in this paradigm. Suppose $\bigcap_P^{t_1} i^{th}$ means the $i^{th}$ instance of the patient over time $t_1$, $\bigcap_{PM}^{t_2} j^{th}$ means the $j^{th}$ instance of paramedical staff (PM) over time $t_2$, and $\bigcap_{CS}^{t_3} k^{th}$ means the $k^{th}$ instance of a cloud server (CS) over time interval $t_3$. The establishment of a partnership and the computation of SK are contingent on meeting specific conditions, which we will now discuss in detail:

1. $\bigcap_P^{t_1} i^{th}$ and $\bigcap_{PM}^{t_2} j^{th}$ or $\bigcap_{PM}^{t_2} j^{th}$ and $\bigcap_{CS}^{t_3} k^{th}$ are in the accepted state

2. $\bigcap_P^{t_1} i^{th}=\bigcap_{PM}^{t_2} j^{th}$ means the partnership between P and PM is not empty.

3. When the **_Reveal(.)_** query is not accepted – it means that the instance is fresh

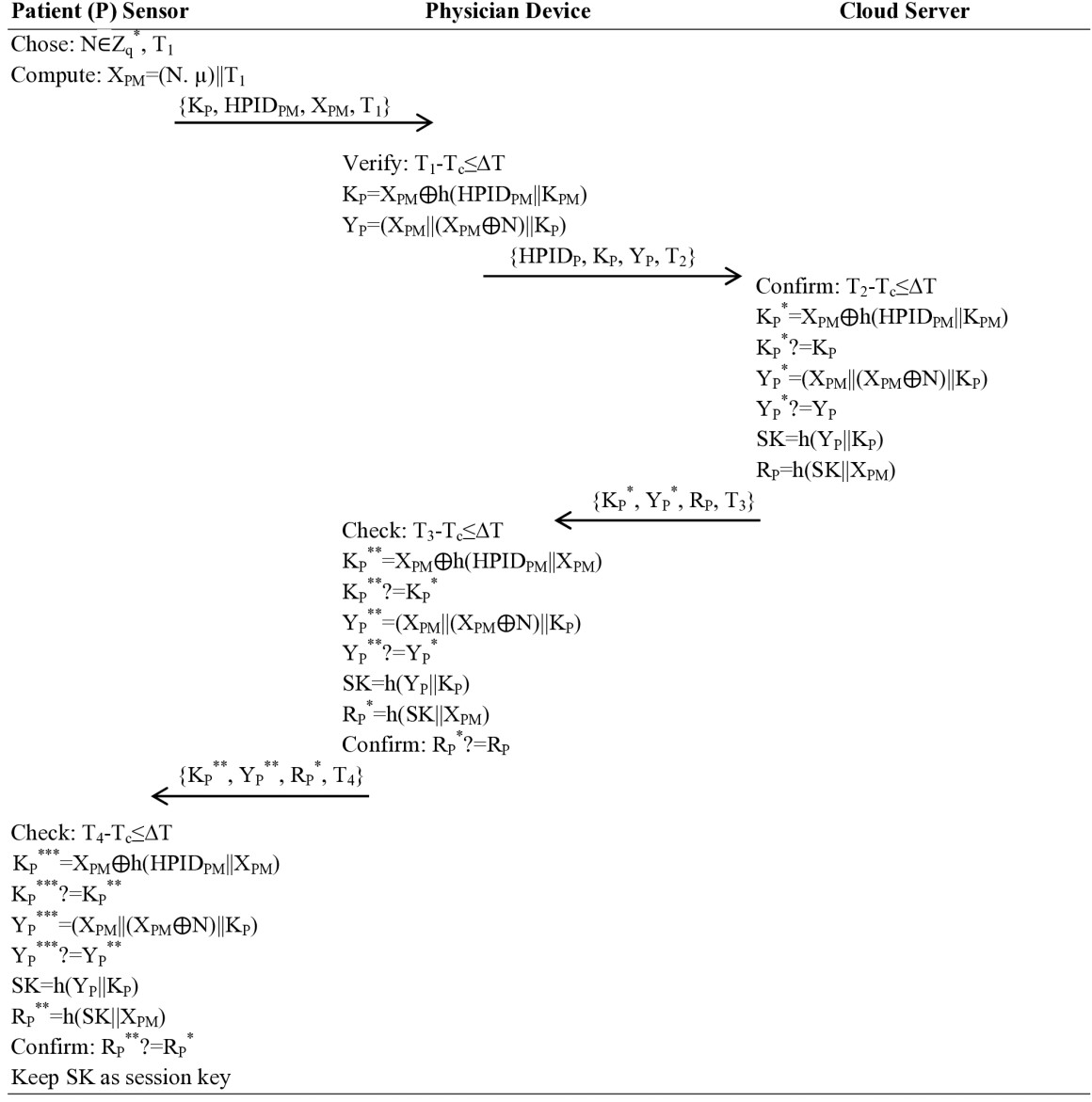

| Patient (P) Sensor | Physician Device | Cloud Server |
|---|---|---|

Chose: $N \in Z_q^*$, $T_1$
Compute: $X_{PM} = (N \cdot \mu) \| T_1$

$\{K_P, HPID_{PM}, X_{PM}, T_1\}$ →

Verify: $T_1 - T_c \leq \Delta T$
$K_P = X_{PM} \oplus h(HPID_{PM} \| K_{PM})$
$Y_P = (X_{PM} \| (X_{PM} \oplus N) \| K_P)$

$\{HPID_P, K_P, Y_P, T_2\}$ →

Confirm: $T_2 - T_c \leq \Delta T$
$K_P^* = X_{PM} \oplus h(HPID_{PM} \| K_{PM})$
$K_P^* ?= K_P$
$Y_P^* = (X_{PM} \| (X_{PM} \oplus N) \| K_P)$
$Y_P^* ?= Y_P$
$SK = h(Y_P \| K_P)$
$R_P = h(SK \| X_{PM})$

← $\{K_P^*, Y_P^*, R_P, T_3\}$

Check: $T_3 - T_c \leq \Delta T$
$K_P^{**} = X_{PM} \oplus h(HPID_{PM} \| X_{PM})$
$K_P^{**} ?= K_P^*$
$Y_P^{**} = (X_{PM} \| (X_{PM} \oplus N) \| K_P)$
$Y_P^{**} ?= Y_P^*$
$SK = h(Y_P \| K_P)$
$R_P^* = h(SK \| X_{PM})$
Confirm: $R_P^* ?= R_P$

← $\{K_P^{**}, Y_P^{**}, R_P^*, T_4\}$

Check: $T_4 - T_c \leq \Delta T$
$K_P^{***} = X_{PM} \oplus h(HPID_{PM} \| X_{PM})$
$K_P^{***} ?= K_P^{**}$
$Y_P^{***} = (X_{PM} \| (X_{PM} \oplus N) \| K_P)$
$Y_P^{***} ?= Y_P^{**}$
$SK = h(Y_P \| K_P)$
$R_P^{**} = h(SK \| X_{PM})$
Confirm: $R_P^{**} ?= R_P^*$
Keep SK as session key

**Fig 4. Authentication and Key Agreement.**

4. The adversary A performs different queries against the proposed protocol as described as follows:

- **Send($\cap^t$, M):** In this query, the adversary A sends a message M towards $\cap^t$ and obtains some output. The adversary A keeps the received output in his record, which is a crucial step in his reconnaissance and potential future attacks.

- **Corrupt($\cap_{PM}^{t_2}$ $j^{th}$):** The adversary A meticulously uses this query to corrupt the mobile device with the paramedical staff and extract secret credentials like s, secret tokens, and the ECC key from memory.

- **Corrupt($\cap_P^{t_1}$ $i^{th}$):** The adversary A employs precise methods in this query to corrupt the embedded sensor or wearable for physiological data collection of the patient's body and extract the secret credentials, such as s, secret tokens, and ECC-key, from memory.

| | ALGORITHM – 1: MUTUAL AUTHENTICATION OF PATIENT, PHYSICIAN AND CLOUD SERVER |
|---|---|
| 1: | Input: $ID_P$, $ID_{PM}$, N, h(.) |
| 2: | Output: SK=h(SK$\|X_{PM}$), and $R_P$=h(SK$\|X_{PM}$) |
| 3: | Extract: $HPID_P$, $HPID_{PM}$, $K_P$, and T |
| 4: | $X_{PM}$=(N. μ)$\|T_1$, μ, s, h(.) |
| 5: | sends {$K_P$, $HPID_{PM}$, $X_{PM}$, $T_1$} message |
| 6: | if($T_1$-$T_c$<=ΔT) then |
| 7: | $K_P$=$X_{PM}$⊕h($HPID_{PM}\|K_{PM}$) |
| 8: | $Y_P$=($X_{PM}\|(X_{PM}$⊕N)$\|K_P$) |
| 9: | sends {$HPID_P$, $K_P$, $Y_P$, $T_2$} message |
| 10: | if($T_2$-$T_c$<=ΔT) then |
| 11: | $K_P{}^*$=$X_{PM}$⊕h($HPID_{PM}\|K_{PM}$) |
| 12: | if($K_P{}^*$==$K_P$) then |
| 13: | $Y_P{}^*$=($X_{PM}\|(X_{PM}$⊕N)$\|K_P$) |
| 14: | if($Y_P{}^*$?=$Y_P$) then |
| 15: | SK=h(SK$\|X_{PM}$) |
| 16: | $R_P$=h(SK$\|X_{PM}$) |
| 17: | sends {$K_P{}^*$, $Y_P{}^*$, $R_P$, $T_3$} |
| 18: | if($T_3$-$T_c$<=ΔT) then |
| 19: | Repeat Step 13 to 16 |
| 20: | sends {$K_P{}^{**}$, $Y_P{}^{**}$, $R_P{}^*$, $T_4$} |
| 21: | if($T_4$-$T_c$<=ΔT) then |
| 22: | Repeat Step 13 to 16 |
| 23: | Return (SK)          Pass |
| 24: | else |
| 25: | Return (0)          Failed |
| 26: | Return (SK)          Pass |
| 27: | else |
| 28: | Return (0)          Failed |
| 29: | Return (SK)          Pass |
| 30: | else |
| 31: | Return (0)          Failed |
| 32: | Return (SK)          Pass |
| 33: | else |
| 34: | Return (0)          Failed |
| 35: | Return (SK)          Pass |
| 36: | else |
| 37: | Return (0)          Failed |
| 38: | Return (SK)          Pass |
| 39: | else |
| 40: | Return (0)          Failed |
| 41: | end if |

**Fig 5. Algorithmic representation of the proposed protocol.**

- **_Execute_**($\bigcap_P^{t_1} i^{th}$, $\bigcap_{PM}^{t_2} j^{th}$, $\bigcap_{CS}^{t_3} k^{th}$): The adversary actively eavesdrops among P, PM, and CS to notice, update, divert the flow, and falsify the publically transmitted credentials.

- **_Reveal_**($\bigcap^t$): In this query, the adversary A discloses the session key SK by acting as a man–in–the–middle between P and PM, PM and CS, CS and PM, or PM and P. This could lead to unauthorized access and potential data breaches, highlighting the severity of the threat.

- **Test($\cap^t$):** After numerous attempts, the adversary A tosses a coin. If the output becomes 1, SK is successfully computed; if the output is zero, the adversary failed; however, if it gets nothing, it means a null value ($\perp$).

5. **Semantic Security:** Suppose W means probability with A in winning a game among a series of games to differentiate among SK and taken under the RoR model. In this regard, adversary A guesses the random number in the Test(.) query. Suppose the guess of A is accurate; then the advantage with A in winning the first game is

$$AD_A^P(t) = \left|2Prob\left[W(A)\right] - 1\right| = |2Prob[R^* = R - 1]|$$

For an actual attack, the advantage with A in winning the game is

$$AD_A^P(t) = |2Prob\left[W_0\right] - 1|$$

For passive attack

$$Prob\left[W_1\right] = Prob\left[W_0\right]$$

According to the birthday paradox [35], the probability of hash images is $\frac{q_h^2}{2^{l_h+1}}$, and random number hash queries are $\frac{(q_e+q_s)^2}{2^{l_r+1}}$, then the advantage with A in polynomial time t for the collision of the hash query is

$$Prob\left[W_2\right] - Prob\left[W_1\right] \le \frac{q_h^2}{2^{l_h+1}} + \frac{(q_e+q_s)^2}{2^{l_r+1}}$$

Next, for obtaining the public key, secret key, and point over curve $K_P$

$$Prob\left[W_3\right] - Prob\left[W_2\right] \le Max\left(\frac{q_s}{2^{l_{ECC}}},\ s,\ K_P\right)$$

Breaking the ECC-Key, the adversary gets

$$Prob\left[W_4\right] - Prob\left[W_3\right] \le AD_A^{ECC}(t)$$

Upon applying the Test(.) Query: adversary gets

$$Prob\left[W_4\right] = \frac{1}{2}$$

Finally,

$$AD_A^P(t) \le \frac{q_h^2}{2^{l_h+1}} + \frac{(q_e+q_s)^2}{2^{l_r+1}} + 2Max\left(\frac{q_s}{2^{l_{ECC}}},\ s,\ K_P\right) + 2AD_A^{ECC}(t)$$

The responses from the various queries and semantic security analysis in the Oracle answer, particularly the encryption of the queried message or a randomly chosen string of the same length, consistently demonstrate the robust security of the proposed security protocol. This should reinforce your sense of security, as adversaries cannot break the hash code, random numbers, or ECC key, further protecting the protocol's security.

## ProVerif simulation

This toolkit simulated a man-in-the-middle attack and determined that the proposed protocol fulfills essential security features, including secrecy, authentication, and process equivalencies. It is capable of handling an infinite message space and

an unlimited number of sessions. The ProVerif software verification toolkit, a renowned tool in the field, automatically transforms the proposed protocol into Horn clauses (a specific type of logical formula). This renowned software verification tool [44] evaluates the protocol for reachability, session key secrecy, and confidentiality, demonstrating that the attacker cannot gain control of the session key at any stage of the authentication process among the involved entities, as shown in Fig 6.

```
Warning: identifier sk rebound.
Completing equations...
-- Process 1-- Query not attacker(skp[]) in process 1
Translating the process into Horn clauses...
Completing...
Starting query not attacker(skp[])
RESULT not attacker(skp[]) is true.
-- Query not attacker(skpm[]) in process 1
Translating the process into Horn clauses...
Completing...
Starting query not attacker(skpm[])
RESULT not attacker(skpm[]) is true.
-- Query not attacker(skcs[]) in process 1
Translating the process into Horn clauses...
Completing...
Starting query not attacker(skcs[])
RESULT not attacker(skcs[]) is true.
-- Query inj-event(UserAuth(idp)) ==> inj-event(UserStart(idp)) in process 1
Translating the process into Horn clauses...
Completing...
Starting query inj-event(UserAuth(idp)) ==> inj-event(UserStart(idp))
RESULT inj-event(UserAuth(idp)) ==> inj-event(UserStart(idp)) is true.

------------------------------------------------------------
Verification summary:

Query not attacker(skp[]) is true.

Query not attacker(skpm[]) is true.

Query not attacker(skcs[]) is true.

Query inj-event(UserAuth(idp)) ==> inj-event(UserStart(idp)) is true.

------------------------------------------------------------
```

**Fig 6. ProVerif Validation – Summary of Verification.**

The code provided includes the definition of two queries, which allows for a thorough examination and the identification of potential loopholes or the simulation of malicious user actions. The above mentioned result summary from this code, a key component of our assessment, demonstrates that the SK remains uncompromised, and an attacker would be unable to exploit it from an open line. This leads to the conclusion that a man-in-the-middle attack, insider attack, forgery attack, and session key hijacking attack are all infeasible on the proposed protocol.

## Pragmatic discussion

In this section of the article, the proposed cloud-assisted key agreement protocol will be informally assessed against various well-known attacks. The discussions about these different attacks are as follows:

**MITM attack.** Suppose the adversary captures the first transmitted message $\{K_P, HPID_{PM}, X_{PM}, T_1\}$ between P and PM. This message consisted of $HPID_P=h(ID_P||N)$, $K_P= N. \mu$, and $X_{PM}=(N. \mu)||T_1$, which means nothing is open or in plain text format, so the adversary failed to act as a malicious user between P and PM. Now, suppose they capture the second transmitted message $\{HPID_P, K_P, Y_P, T_2\}$ which consisted of $HPID_P=h(ID_P||N)$, $Y_P=(X_{PM}||(X_{PM}\oplus N)||K_P)$ and $K_P=X_{PM}\oplus h(HPID_{PM}||K_{PM})$ which means nothing is open, all the credentials are not in plain text format, so adversary cannot find anything helpful to act as MITM. Therefore, the proposed protocol resists man-in-the-middle attacks.

**Key disclosure attack.** If an attacker captures the SK from the memory of either SN or a mobile device and desires to figure out something useful from it, as $SK=h(Y_P||K_P)$, which means $Y_P=(X_{PM}||(X_{PM}\oplus N)||K_P)$ and $K_P=X_{PM}\oplus h(HPID_{PM}||K_{PM})$, they couldn't find anything useful or in plain text format. Therefore, a key disclosure attack is not possible on the proposed protocol.

**Traceability attack.** If the attacker attempts to trace out the patient or physician or the information, due to 160 bits of ECC key, SHA256 hash function, and complex set of calculations, privacy is preserved for all the entities. The physician received $\{K_P, HPID_{PM}, X_{PM}, T_1\}$ message containing $HPID_P=h(ID_P||N)$, $K_P= N. \mu$, and $X_{PM}=(N. \mu)||T_1$, the cloud server receiving $\{HPID_P, K_P, Y_P, T_2\}$ consisted of $HPID_P=h(ID_P||N)$, $K_P=X_{PM}\oplus h(HPID_{PM}||K_{PM})$, and $Y_P=(X_{PM}||(X_{PM}\oplus N)||K_P)$, and the patient side received $\{K_P^{**}, Y_P^{**}, R_P^*, T_4\}$ parameters having $K_P^*=X_{PM}\oplus h(HPID_{PM}||K_{PM})$, and $Y_P^*=(X_{PM}||(X_{PM}\oplus N)||K_P)$ which means nothing is concealed to the attacker. So, traceability attacks are not valid for the proposed protocol.

**Insider threat.** The sensor node embedded inside the patient body or wearable for other vital collection consisted of $\{s, X_P, H_P, K_P\}$ parameters; if the illegitimate one attempts to calculate some parameters from it, they will fail because of $K_P=N.\mu$ and $H_P=h(ID_P||K_P)$, $X_P=(N\oplus H_P)||s$ so adversary was unable to act as an insider. Similarly, the memory of a mobile device consists of $\{s, X_{PM}, H_{PM}, K_P\}$, which is a set of complex calculations; again, the adversary cannot act as an insider, and the memory of the cloud server also has nothing in plain text format, so the adversary cannot succeed for launching an insider threat on any of the entity. Therefore, an insider attack on the proposed protocol is not possible.

**DoS attack.** The patient side calculates $X_{PM}=(N. \mu)||T_1$ and sends $\{K_P, HPID_{PM}, X_{PM}, T_1\}$ before further computation. The doctor-side IoT device or mobile phone verifies the time $T_1-T_c\leq\Delta T$ and sends $\{HPID_P, K_P, Y_P, T_2\}$ message towards the cloud server. The cloud server, as the final gatekeeper, verifies the time threshold $T_2-T_c\leq\Delta T$, computes, $K_P^*=X_{PM}\oplus h(HPID_{PM}||K_{PM})$, confirms $K_P^*?=K_P$, $Y_P^*=(X_{PM}||(X_{PM}\oplus N)||K_P)$, and confirms $Y_P^*?=Y_P$ to pass to the next step. This crucial role of the cloud server in the verification process ensures the mitigation of both DoS and DDoS attacks, providing a strong sense of reassurance in the proposed protocol's robustness.

**Replay attack.** Verification of time with the current system, along with the parameters $K_P^*?=K_P$, $Y_P^*?=Y_P$ in the first round, and $K_P^{**}?=K_P^*$, $Y_P^{**}?=Y_P^*$ in the second round, plays a crucial role in the mitigation of replay attacks in the proposed protocol. The system's refusal to allow replay attacks is further reinforced by the computation steps, $SK=h(Y_P||K_P)$, $R_P=h(SK||X_{PM})$, $Y_P^{**}=(X_{PM}||(X_{PM}\oplus N)||K_P)$, $K_P^{***}=X_{PM}\oplus h(HPID_{PM}||X_{PM})$.

**Tracking attack.** An attacker's attempts to track the original identity, keys, or sensitive credentials are futile, even if they manage to obtain valuable information through the public network channel. This is due to our robust security measures, which include using ECC keys and 60-bit random numbers that are dynamically changed after each proposed

key agreement protocol round-trip. This approach not only enhances security but also ensures that the patient/physician cannot be tracked by the adversary at any stage of the communication. It's reassuring that tracking the position of the patient, physician, or cloud server remains secure, thereby providing a high level of trust in the effectiveness of our recommended security measures and showing resilience to a tracking attack.

**Desynchronization attack.** The proposed protocol, involving both the sensor and the physician, is designed to handle desynchronization. It is a reliable system that can adapt to various situations and maintain communication. From the physician's end to the patient, the synchronization characteristics, $PK_P$, and $PK_{MP}$ are modified before being replaced with distinctive features. Even if an insider changes some credentials or floods the cloud server with new credentials, the protocol ensures the communication's integrity by preserving the available data, maintain the synchrony of shared secret thereby providing a secure environment for communication to all the participating entities in efficient and effective manner.

## Performance analysis

In this section of the article, a comprehensive analysis of the proposed protocol's metrics, including storage, communication, and computation costs, are thoroughly examined. Following this, the proposed key agreement protocol will be compared with the state-of-the-art schemes in terms of communication and computation costs and then measured the energy consumption of it. This meticulous approach ensures the validity and reliability of the proposed key agreement scheme. So, these different analyses are described one by one as follows:

### Computation cost analysis

For the implementation of the proposed protocol, the MIRACLE crypto SDK [45] was utilized, employing the C programming language across a diverse range of three devices: the Raspberry Pi 5, which is powered by a Broadcom BCM2712 quad-core Arm Cortex A76 processor running at 2.4GHz; the Samsung Galaxy A05s featuring an octa-core processor (2x2.0 GHz Cortex-A75 & 6x1.8 GHz Cortex-A55) with 6GB of RAM; and a laptop equipped with a Core i7-6500U processor at 2.5 GHz and 16GB of RAM. The execution times for various cryptographic operations, detailed in Table 3, offer valuable insights into the performance capabilities of each device. Notably, the execution time for the Raspberry Pi 5 is considered in the context of a patient's sensor/wearable device, the Samsung Galaxy A05s serves as the mobile device for paramedical staff, and the Core i7 laptop functions as the cloud server.

So far, for the proposed protocol, the number of multiplication operations is $5T_X$, the number of addition operations is $8T_+$ and the number of hash cryptographic operations is $10T_H$, resulting in a precise computation cost of $5T_X + 8T_+ + 10T_H = 5(1.67) + 8(1.52) + 10(2.47) = 8.35 + 12.16 + 24.7 = 45.21$ ms for the sensor/wearable for patient devices. When measured for a doctor, nurse, or paramedical staff device, the cost is $=5(0.68) + 8(0.98) + 10(1.11) = 3.4 + 7.84 + 11.1 = 22.34$ ms. finally, for the cloud server, the cost is $=5(0.91) + 8(0.78) + 10(0.91) = 4.55 + 6.24 + 9.1 = 19.89$ ms, as shown in Table 4 and plotted in Fig 7.

**Table 3. Execution Time for Different Cryptographic Operations.**

| Operation Name | Symbol | Raspberry Pi 5 (P) | Samsung Galaxy A05s (PM) | Core i7 Laptop (CS) |
|---|---|---|---|---|
| Hash Function | $T_H$ | 2.47 | 1.11 | 0.91 |
| Multiplication Function | $T_X$ | 1.67 | 0.68 | 0.43 |
| Addition Function | $T_+$ | 1.52 | 0.98 | 0.78 |

**Table 4. Computation Cost in Milliseconds.**

| Participant | Total Cost |
|---|---|
| Patient (P) | 45.21 |
| Doctor (PM) | 22.34 |
| Cloud Server (CS) | 19.89 |
| **Total Cost in Milliseconds** | **87.44 ms** |

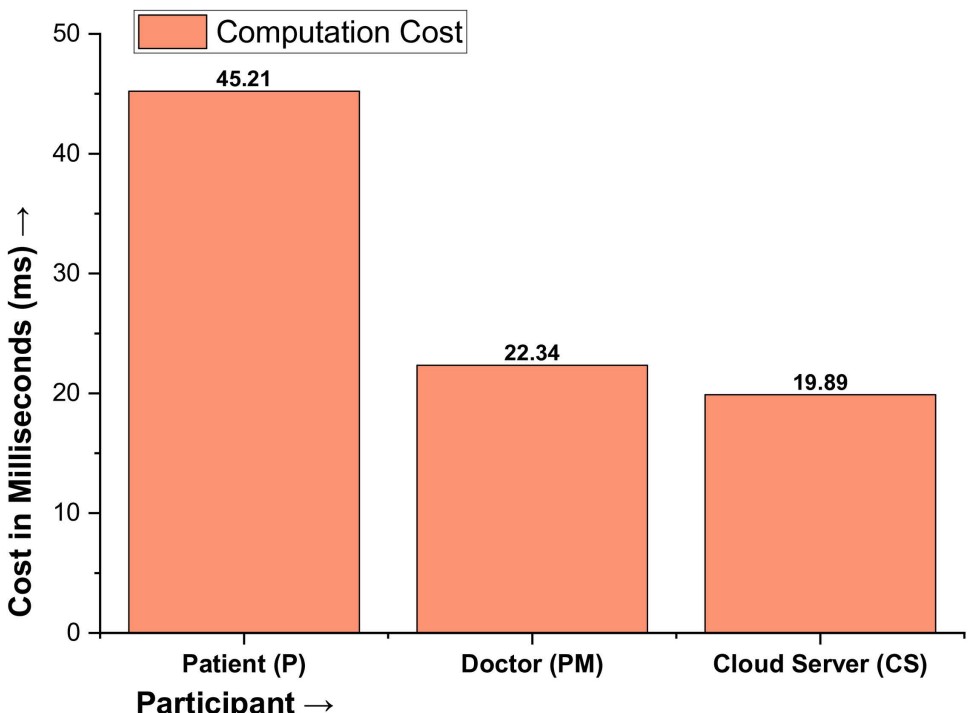

**Fig 7. Computation Cost of the Proposed Protocol.**

## Communication cost analysis

The communication cost is basically the bandwidth utilized during the exchange of various parameters and messages in the key agreement phase of the protocol. According to [38,46] the hash function occupies a space of 256 bits because, using the SHA256 algorithm, random numbers are 60-bit, a 64-bit identity, a 160-bit ECC key, and a 32-bit timestamp. So, the first transmitted message {$K_P$, $HPID_{PM}$, $X_{PM}$, $T_1$}, incurs a total cost of 704 bits, calculated as 160 + 256 + 256 + 32 = 704; the second message, {$HPID_P$, $K_P$, $Y_P$, T2}, costs 800 bits, with a breakdown of 256 + 256 + 256 + 32 = 800 bits, and the third message, {$K_P^*$, $Y_P^*$, $R_P$, $T_3$}, also cost 800 bits, following the same breakdown; while, the last message, {$K_P^{**}$, $Y_P^{**}$, $R_P^*$, $T_4$}, again total 800 bits. The overall communication cost of the proposed protocol is 3104 bits, calculated as 704 + 800 + 800 + 800 = 3104 bits, as highlighted in Table 5 and illustrated in Fig 8.

## Storage cost analysis

The memory of the patient sensor stores parameters {s, $X_P$, $H_P$, $K_P$} with a total cost of 64 + 256 + 256 + 160 = 736 bits (according to [46]). The doctor device memory consists of parameters {s, $X_{PM}$, $H_{PM}$, $K_P$}, which also amounts to 64 + 256 +

**Table 5. Communication cost of the proposed protocol in bits.**

| Participant | Message | Values | Cost |
|---|---|---|---|
| P →PM | $\{K_P, HPID_{PM}, X_{PM}, T_1\}$ | 160 + 256 + 256 + 32 | 704 |
| PM →CS | $\{HPID_P, K_P, Y_P, T_2\}$ | 256 + 256 + 256 + 32 | 800 |
| CS →PM | $\{K_P^*, Y_P^*, R_P, T_3\}$ | 256 + 256 + 256 + 32 | 800 |
| PM →P | $\{K_P^{**}, Y_P^{**}, R_P^*, T_4\}$ | 256 + 256 + 256 + 32 | 800 |
| **Total Communication cost in Bits** | | | **3104 bits** |

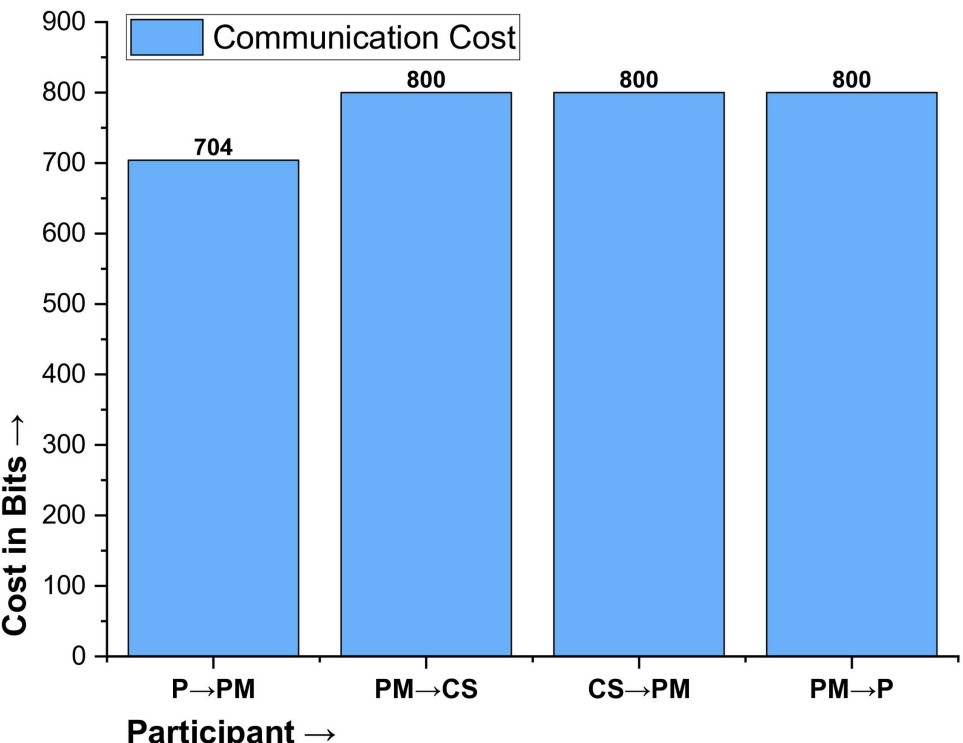

**Fig 8. Communication Cost of the Proposed Protocol.**

256 + 160 = 736 bits (according to [46]). The cloud server memory includes i) Parameters $\{s, \mu, h(.), E_q(a, b)\}$ with a cost of 64 + 32 + 56 + 256 = 408 bits, ii) Parameters $\{s, X_P, H_P, K_P\}$ costing 64 + 256 + 256 + 160 = 736 bits, and iii) the parameters $\{s, X_{PM}, H_{PM}, K_P\}$ also costing 64 + 256 + 256 + 160 = 736 bits. The cloud server stores 408 + 736 + 736 = 1880 bits. Thus, the overall storage cost of the proposed protocol is calculated as follows: 736 + 736 + 1880 = 3352 bits, as detailed in Table 6 and illustrated in Fig. 9.

## Comparative analysis

When comparing the proposed protocol with Mohit et al. [47], Li et al. [48], Sahoo et al. [49], and Zhou et al. [50] in terms of performance metrics (communication and computation costs), the result in Table 7 provides a comprehensive comparison. It demonstrates that the proposed key agreement protocol is lightweight and robust, as shown diagrammatically in Fig 10, and outperforms the existing protocols in these aspects.

                                                                

**Table 6. Storage cost of the proposed protocol in bits.**

| Entity | Parameters | Value | Cost |
|---|---|---|---|
| Sensor Memory | {s, $X_P$, $H_P$, $K_P$} | 64 + 256 + 256 + 160 | 736 |
| Device Memory | {s, $X_{PM}$, $H_{PM}$, $K_P$} | 64 + 256 + 256 + 160 | 736 |
| Cloud Server Memory | {s, μ, h(.), $E_q$(a, b)} <br> {s, $X_P$, $H_P$, $K_P$} <br> {s, $X_{PM}$, $H_{PM}$, $K_P$} | 64 + 32 + 56 + 256 <br> 64 + 256 + 256 + 160 <br> 64 + 256 + 256 + 160 | 1880 |
| **Total Storage costs in Bits** | | | **3352 bits** |

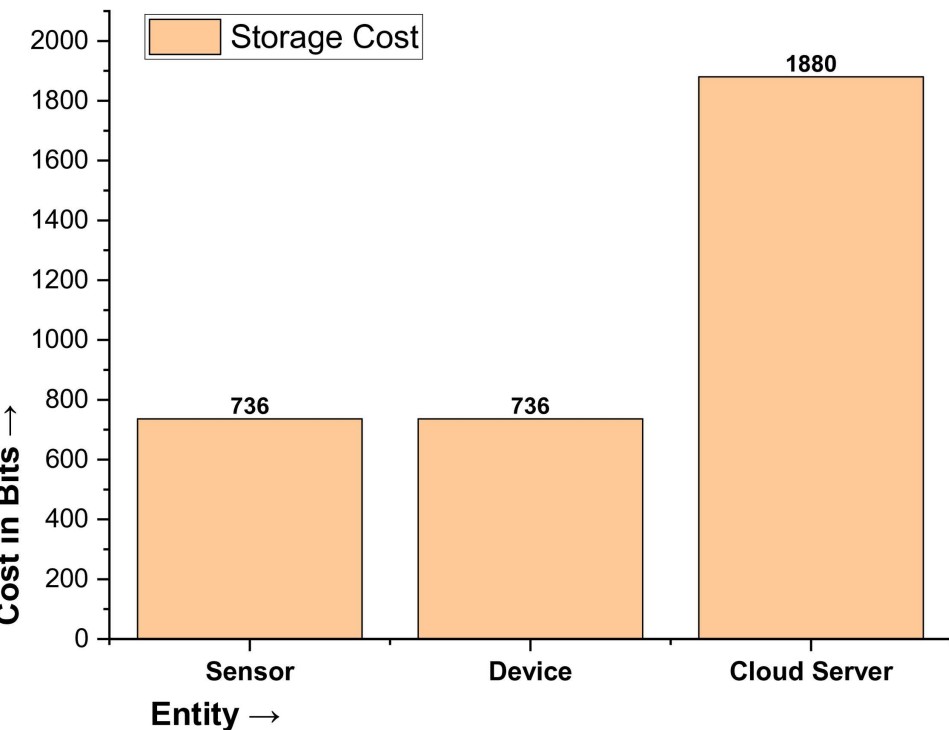

**Fig 9. Storage Cost of the Proposed Protocol.**

**Table 7. Comparative Analysis.**

| Protocols → <br> Performance Metrics↓ | [47] | [48] | [49] | [50] | [Proposed] |
|---|---|---|---|---|---|
| Communication Cost in Bits | 5312 | 4096 | 4332 | 5856 | 3104 |
| Computation Cost in Milliseconds (ms) | 208.6 | 2470.4 | 96.34 | 111.35 | 87.44 |
| Energy Consumption (J) | 2.3 | 26.9 | 1.5 | 1.3 | 0.96 |

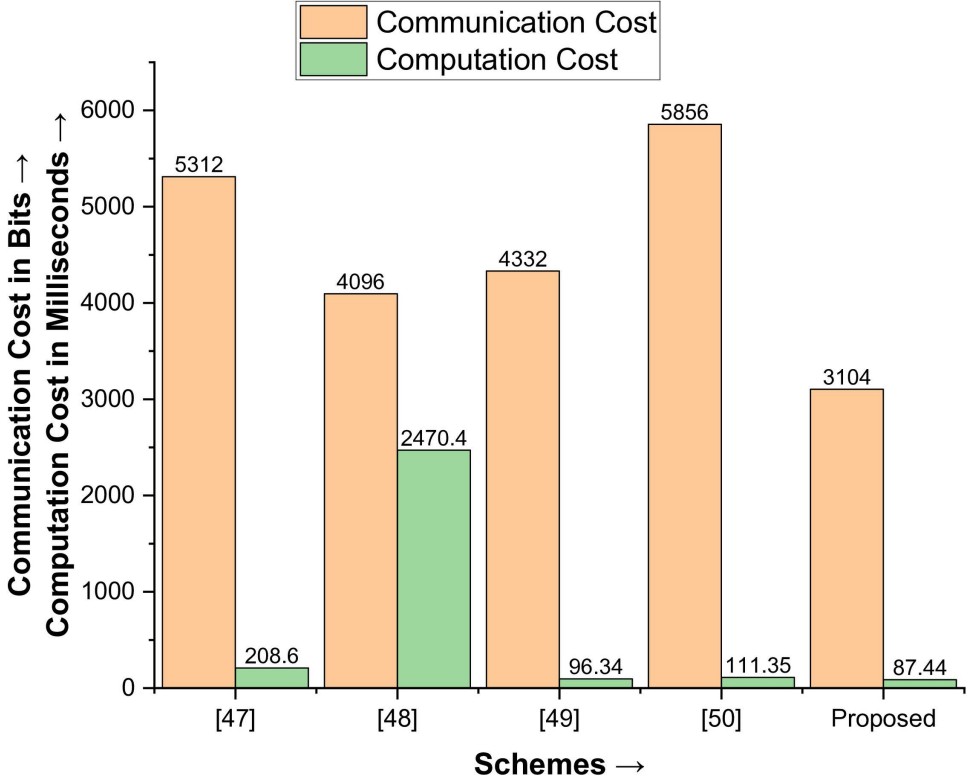

**Fig 10. Comparative Analysis – A diagrammatic overview.**

The proposed protocol stands out with its exceptional performance. It is 41.57% better in terms of communication cost than Mohit et al. [47], 24.22% from Li et al. [48], 28.35% from Sahoo et al. [49], and 46.99% from Zhou et al. [50]. Similarly, it is 58.08% better in terms of computation costs from Mohit et al. [47], 96.46% from Li et al. [48], 9.35% from Sahoo et al. [49], and 21.47% from Zhou et al. [50]. These results underscore the overall outstanding performance of the proposed protocol.

## Energy consumption

The resources use a certain amount of battery power while the proposed key agreement protocol is executed as soon as its initialization phase is run. The amount of power consumed by the Raspberry Pi, laptop, and cell phone for calculating the session secret key is represented by the equation $E_x = C_Y \times C_Z$ for a wireless channel [51]. The reliability of the proposed key agreement protocol is evident in its consistent power consumption. Suppose the computation costs of the proposed key agreement protocol is $C_Y$, which is 87.44 ms, and suppose $C_Z$ is the CPU's maximum power consumption, which is fixed and equal to 10.88 Watts, according to [52–53] for wireless data transmission. By entering these numbers into the calculation, $E_x$ = 87.44 x 10.88 = 951.34 mJ or 0.96 joule. Therefore, 0.88-joule power is consumed upon running the proposed key agreement protocol for security authentication all the participating entities in the cloud computing paradigm and computes session secret key SK.

## Conclusion

This research introduces a cloud-assisted key agreement protocol designed for secure communication within the e-healthcare system, utilizing ECC and SHA256 cryptographic primitives. The proposed method facilitates

cross-verification among the parties involved and ensures the legality of participating entities in the cloud computing paradigm. What sets this protocol apart is the thoroughness of its examination, which includes formal security proofs utilizing the RoR model and ProVerif simulation, as well as informal security proofs through pragmatic illustrations. This comprehensive approach has been evaluated by considering computation, communication, storage costs, and energy consumption metrics, with results indicating that it is lightweight, effective for cloud setting, robust against known threats, and fulfills essential security requirements. Comparisons with state-of-the-art schemes reveal that the proposed protocol outperforms others regarding communication and computation costs, thus affirming its applicability in the real-world scenario. In future, the same e-healthcare setting can be secured by designing a protocol using advanced AI and machine learning algorithms and can be explained with the help of a real-world use case.

## Supporting Information

**S2 File 1. ProVerif Simulation Code.**
(TXT)

## Acknowledgement

The author would like to thank AlMaarefa University, Riyadh, Saudi Arabia, for continuous support and encouragement.

## Author contributions

**Conceptualization:** Ismail Keshta.

**Data curation:** Ismail Keshta.

**Formal analysis:** Ismail Keshta.

**Funding acquisition:** Ismail Keshta.

**Investigation:** Ismail Keshta.

**Methodology:** Ismail Keshta.

**Project administration:** Ismail Keshta.

**Resources:** Ismail Keshta.

**Software:** Ismail Keshta.

**Supervision:** Ismail Keshta.

**Validation:** Ismail Keshta.

**Visualization:** Ismail Keshta.

**Writing – original draft:** Ismail Keshta.

**Writing – review & editing:** Ismail Keshta.

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
