## [Decision Letter · Decision Letter 0]

12 Feb 2025

PONE-D-24-60331A Cloud-Assisted Key Agreement Protocol for the E-Healthcare SystemPLOS ONE

Dear Dr. Keshta,

Thank you for submitting your manuscript to PLOS ONE. After careful consideration, we feel that it has merit but does not fully meet PLOS ONE’s publication criteria as it currently stands. Therefore, we invite you to submit a revised version of the manuscript that addresses the points raised during the review process.

We look forward to receiving your revised manuscript.

Kind regards,

Arijit Karati

Academic Editor

PLOS ONE

Journal Requirements:

**Additional Editor Comments:**

The work has some potential benefits in the secure key agreement. However, the reviewers have provided several important comments that should be incorporated in the revised manuscript.

Reviewers' comments:

Reviewer's Responses to Questions

**Comments to the Author**

1. Is the manuscript technically sound, and do the data support the conclusions?

Reviewer #1: Yes

Reviewer #2: Partly

2. Has the statistical analysis been performed appropriately and rigorously? 

Reviewer #1: Yes

Reviewer #2: No

3. Have the authors made all data underlying the findings in their manuscript fully available?

Reviewer #1: Yes

Reviewer #2: No

4. Is the manuscript presented in an intelligible fashion and written in standard English?

Reviewer #1: Yes

Reviewer #2: No

5. Review Comments to the Author

Reviewer #1: 1- The abstract did not clarify the criteria that the researcher adopted to show that the proposed method is better than others, nor the numerical values of the most important of these criteria.

2- The keywords are general and do not contain any specification.

3- The researcher's contributions are very general and the distinctive contribution made by the researcher in this manuscript is not specified. It is possible to include the contributions as a paragraph in the introduction. In addition, the introduction does not contain a paragraph explaining the structure of the research in general.

4- The research paper lacks much clarification through figures or diagrams, in addition to the lack of a discussion that clarifies the researcher’s reading of the results he obtained and comparing them with other previous works.

5- The conclusions are not based on numerical values of the results and the proposed future work is not clearly explained.

Reviewer #2: 1) Towards the end of the abstract, give a summary of the key findings of this study. This may include the security threats prevented by your proposed scheme as well as the computation costs, communication costs, storage overheads, and energy consumption.

2) The quality of English used in this paper must be greatly enhanced.

3) Rewrite the introduction section so that the problem domain in vividly clear to the readers.

4) The motivation section is not well written since you have failed to describe the shortcomings of the current mutual authentication and key agreement protocols.

5) Your contributions do not reflect any novelty: ECC and SHA-256 are well-known cryptographic techniques. What is so unique about your proposed scheme that differentiates it from previous ECC-based and SHA-256 based schemes?

6) Your third contribution reads as follows: "The proposed protocol's performance is tested by measuring computation, communication, strong overheads, and energy consumption, ensuring its practicality and reliability for real-world implementation."

---Replace 'strong overheads' with 'storage overheads'. Proof-read the entire paper to eliminate similar grammatical and spelling mistakes.

7) Before introducing the Related works, add a paragraph to describe how this paper is organized.

8) In the related works section, do the following:

i ) Describe the techniques used in the related works, as well as the pros and cons of all presented works

ii) Add a table to summarize the presented works in terms of the techniques deployed, pros and cons

iii) After the table in (ii) above, add a paragraph to summarize all the identified research gaps. Thereafter, describe how your proposed solution helps alleviate these challenges.

9) For enhanced readability, the various phases of your proposed protocol must be presented in a step-wise manner.

10) Under the 'Proposed Protocol' section, you have claimed that the proposed scheme included setup, registration, and key agreement phases. However, the abstract indicated that you are developing a cloud assisted key agreement protocol for the e-healthcare system to enable secure authentication for patient monitoring, enhancing mutual authentication of the participating entities and creating secure session key among all them for secure open channel communication.

--- Where is the mutual authentication aspect of your protocol? Is it not supposed to be one of the key phases in your proposed protocol?

11) Use proper names to name your algorithms and figures. You have used Module 1 and Module 2, which are quite confusing.

12) Under 'ProVerif Simulation', you claim that This toolkit simulated the launch of a man-in-the-middle attack and found that the proposed protocol meets the needed security features, such as secrecy, authentication, and process equivalencies.

-- How can readers verify the claim you make above from the ProVerif output?

--You need to describe the output of the ProVerif output.

13) Under the 'Pragmatic Discussion', add more attack vectors

14) In your third contribution, you claimed to have analyzed your proposed protocol in terms of energy consumption. However, this has not been presented under the 'Performance Analysis' section. You must present this energy consumption analysis.

15) The greatest weakness of the 'Performance Analysis' section is that no comparative evaluation has been carried out. You must compare the performance of your protocol against the state-of-the-art protocols in this domain. In addition, you must describe and interpret all the obtained results.

16) The conclusion section is poorly done and it fails to capture the key findings, practical implications of the obtained results, limitations of your proposed protocol, as well as future research directions.

6. PLOS authors have the option to publish the peer review history of their article (what does this mean? ). If published, this will include your full peer review and any attached files.

**Do you want your identity to be public for this peer review?** For information about this choice, including consent withdrawal, please see our Privacy Policy .

Reviewer #1: No

Reviewer #2: No

---

## [Author Response · Author response to Decision Letter 1]

17 Feb 2025

I dedicated to addressing all the reviewers' concerns and have worked diligently to enhance the scholarly quality of the manuscript. The reviewers pointed out several significant issues, which I have thoroughly addressed to the best of my ability.

All the concerns are fully addressed and make the articel worth scholar....

---

## [Decision Letter · Decision Letter 1]

28 Feb 2025

PONE-D-24-60331R1A Cloud-Assisted Key Agreement Protocol for the E-Healthcare SystemPLOS ONE

Dear Dr. Keshta,

Thank you for submitting your manuscript to PLOS ONE. After careful consideration, we feel that it has merit but does not fully meet PLOS ONE’s publication criteria as it currently stands. Therefore, we invite you to submit a revised version of the manuscript that addresses the points raised during the review process.

**The work has been revised based on the earlier comments. However, there are a few critical comments that should be incorporated.**

We look forward to receiving your revised manuscript.

Kind regards,

Arijit Karati

Academic Editor

PLOS ONE

**Journal Requirements:**

**Additional Editor Comments:**

The work has been revised based on the earlier comments. However, there are a few critical comments that should be incorporated.

Reviewers' comments:

Reviewer's Responses to Questions

**Comments to the Author**

1. If the authors have adequately addressed your comments raised in a previous round of review and you feel that this manuscript is now acceptable for publication, you may indicate that here to bypass the “Comments to the Author” section, enter your conflict of interest statement in the “Confidential to Editor” section, and submit your "Accept" recommendation.

Reviewer #1: All comments have been addressed

Reviewer #2: (No Response)

2. Is the manuscript technically sound, and do the data support the conclusions?

Reviewer #1: (No Response)

Reviewer #2: Partly

3. Has the statistical analysis been performed appropriately and rigorously? 

Reviewer #1: (No Response)

Reviewer #2: N/A

4. Have the authors made all data underlying the findings in their manuscript fully available?

Reviewer #1: (No Response)

Reviewer #2: Yes

5. Is the manuscript presented in an intelligible fashion and written in standard English?

Reviewer #1: (No Response)

Reviewer #2: No

6. Review Comments to the Author

**Reviewer #1: ** (No Response)

**Reviewer #2:**  Thank you for addressing some of the previous comments. However, the following issues still require your attention:

1) The quality of English used in this paper is still low. For instance:

"Lopes et al. [12] proposed a protocol to provide safe and reciprocal device authentication within the system, vulnerable to traceability attacks."

---but is vulnerable to traceability attacks..

"and 53.69% in energy consumption, which is installing its superiority over its competitors."

--'installing' is not appropriate in the above sentence.

2) In the abstract, you have still failed to state some of the security threats prevented by your proposed scheme.

3) In the motivation section, describe some of the shortcomings of the current mutual authentication and key agreement protocols.

4) For each of your research contributions, vividly bring out the uniqueness/novelty compared with other related state-of-the-art solutions.

5) In the Related works section, you have attempted to point out the weaknesses of the current works. However, you have not provided the rationale for your critique. For instance, what makes you believe that the scheme in [12] is vulnerable to traceability attacks? Why do you think that the protocol in [13] has design issues?

--You must either cite or give rationale for the critique provided. This increases the credibility to the identified research gaps in the related works.

6) Replace 'Key Agreement Phase' with 'Authentication and Key Agreement Phase'

7) My previous COMMENT # 11 was as follows:

"Use proper names to name your algorithms and figures. You have used Module 1 and Module 2, which are quite confusing."

--You have indicated that you have made corrections above. However, in Registration section, you have still written as follows:

>>The system ensures the security of the communication by sending {s, XP, HP, KP} back to the patient terminal, where it is stored in its memory, as shown in module 1."

>> Module 1. Patient Registration

>>This data is then stored in the cloud server's memory, as shown in module 2, ensuring the highest level of security.

>>Module 2. Physician Registration

--There are so many instances where you still use 'Module' instead of appropriate names such as 'Figure'. Ensure that all changes you effect are reflected in the entire paper.

8) In Table 7, you have provided some comparative analysis in terms of computation and communication costs. However, you have failed to include storage costs and energy consumption in this Table 7.

7. PLOS authors have the option to publish the peer review history of their article (what does this mean? ). If published, this will include your full peer review and any attached files.

**Do you want your identity to be public for this peer review?** For information about this choice, including consent withdrawal, please see our Privacy Policy .

Reviewer #1: No

Reviewer #2: No

---

## [Author Response · Author response to Decision Letter 2]

1 Mar 2025

A detailed rebuttal letter is attached

---

## [Decision Letter · Decision Letter 2]

18 Mar 2025

A Cloud-Assisted Key Agreement Protocol for the E-Healthcare System

PONE-D-24-60331R2

Dear Dr. Keshta,

We’re pleased to inform you that your manuscript has been judged scientifically suitable for publication and will be formally accepted for publication once it meets all outstanding technical requirements.

Kind regards,

Arijit Karati

Academic Editor

PLOS ONE

Additional Editor Comments (optional):

The manuscript has been revised; however, additional comments must be incorporated into the final version.

Reviewers' comments:

Reviewer's Responses to Questions

**Comments to the Author**

1. If the authors have adequately addressed your comments raised in a previous round of review and you feel that this manuscript is now acceptable for publication, you may indicate that here to bypass the “Comments to the Author” section, enter your conflict of interest statement in the “Confidential to Editor” section, and submit your "Accept" recommendation.

Reviewer #2: (No Response)

2. Is the manuscript technically sound, and do the data support the conclusions?

Reviewer #2: Partly

3. Has the statistical analysis been performed appropriately and rigorously? 

Reviewer #2: N/A

4. Have the authors made all data underlying the findings in their manuscript fully available?

Reviewer #2: Yes

5. Is the manuscript presented in an intelligible fashion and written in standard English?

Reviewer #2: No

6. Review Comments to the Author

Reviewer #2: Thank you for attending to some of the previous comments. Please note the following:

1) Revise the entire document so that the quality of English used is acceptable. Consider the following sentence appearing in your abstract:

"The result obtained from the security analysis demonstrated that the proposed protocol resisting man-in-the-middle, replay, DoS, traceability/tracking, desynchronization, impersonation, and side channel attacks offers key secrecy, confidentiality, integrity, and authorization."

--Revise the above sentence as follows:

"The result obtained from the security analysis demonstrated that the proposed protocol resists man-in-the-middle, replay, DoS, traceability/tracking, desynchronization, impersonation, and side channel attacks. It also offers key secrecy, confidentiality, integrity, and authorization."

"In contrast, the result depicted from the performance analysis section shows that the proposed protocol is 46.99% better in communication, 96.46% in computation, and 53.69% in energy consumption, which is inaugurating its superiority over its competitors."

--Revise the above sentence as follows:

"In contrast, the performance analysis results show that the proposed protocol is 46.99% better in communication, 96.46% in computation, and 53.69% in energy

consumption. This demonstrates its superiority over its peers.

>> There are many of such sentences which needs to be corrected.

2) My previous COMMENT # 5 read as follows:

"In the Related works section, you have attempted to point out the weaknesses of the current works. However, you have not provided the rationale for your critique. For instance, what makes you believe that the scheme in [12] is vulnerable to traceability attacks? Why do you think that the protocol in [13] has design issues?

--You must either cite or give rationale for the critique provided. This increases the credibility to the identified research gaps in the related works."

>> In your response, you have indicated that "Dear Sir, The proof you are demanding occupies too much space. For instance, I am analyzing once again to confirm it, but I assure you that after conducting a thorough investigation, I have identified and critiqued the loopholes and mentioned them. It is essential to note that only

individuals with the proper knowledge can mention the criticism of someone's work."

-- The above response is inaccurate. In research work, you must have rationale for the claims made. For instance, if you claim that the scheme in [12] is vulnerable to traceability attacks, you must justify this claim. For instance, if the scheme in [12] exchanges real identities of the network elements in plain-text, then it is vulnerable to traceability attacks. On the other hand, if the protocol in [13] executes authentication before registration, you can claim that it has claim issues. These are the justifications that you are required to include. I don't see how this justification/rationale occupies too much space.

3) Under comparative analysis, you have written as follows:

"When comparing the proposed protocol with Mohit et al. [47], Li et al. [48], Sahoo et al. [49], and Zhou et al. [50] in terms of performance metrics (communication and computation costs), the result in Table 7 provides a comprehensive comparison."

-- Table 7 now included energy consumption. So revise the above statement as follows:

"When comparing the proposed protocol with Mohit et al. [47], Li et al. [48], Sahoo et al. [49], and Zhou et al. [50] in terms of performance metrics (energy consumption, communication and computation costs), the result in Table 7 provides a comprehensive comparison."

7. PLOS authors have the option to publish the peer review history of their article (what does this mean? ). If published, this will include your full peer review and any attached files.

**Do you want your identity to be public for this peer review?** For information about this choice, including consent withdrawal, please see our Privacy Policy .

Reviewer #2: No

---

## [Editor Report · Acceptance letter]

PONE-D-24-60331R2

PLOS ONE

Dear Dr. Keshta,

I'm pleased to inform you that your manuscript has been deemed suitable for publication in PLOS ONE. Congratulations! Your manuscript is now being handed over to our production team.

Kind regards,

on behalf of

Dr. Arijit Karati

Academic Editor

PLOS ONE
